# A Primal-Dual link between GANs and Autoencoders

**Hisham Husain**‡,†    **Richard Nock**†,‡,♣    **Robert C. Williamson**‡,†

‡The Australian National University, †Data61, ♣The University of Sydney

firstname.lastname@{data61.csiro.au,anu.edu.au}

## Abstract

Since the introduction of Generative Adversarial Networks (GANs) and Variational Autoencoders (VAE), the literature on generative modelling has witnessed an overwhelming resurgence. The impressive, yet elusive empirical performance of GANs has lead to the rise of many GAN-VAE hybrids, with the hopes of GAN level performance and additional benefits of VAE, such as an encoder for feature reduction, which is not offered by GANs. Recently, the Wasserstein Autoencoder (WAE) was proposed, achieving performance similar to that of GANs, yet it is still unclear whether the two are fundamentally different or can be further improved into a unified model. In this work, we study the $f$-GAN and WAE models and make two main discoveries. First, we find that the $f$-GAN and WAE objectives partake in a primal-dual relationship and are equivalent under some assumptions, which then allows us to explicate the success of WAE. Second, the equivalence result allows us to, for the first time, prove generalization bounds for Autoencoder models, which is a pertinent problem when it comes to theoretical analyses of generative models. Furthermore, we show that the WAE objective is related to other statistical quantities such as the $f$-divergence and in particular, upper bounded by the Wasserstein distance, which then allows us to tap into existing efficient (regularized) optimal transport solvers. Our findings thus present the first primal-dual relationship between GANs and Autoencoder models, comment on generalization abilities and make a step towards unifying these models.

## 1 Introduction

Implicit probabilistic models [1] are defined to be the pushforward of a simple distribution $P_Z$ over a latent space $\mathcal{Z}$ through a map $G : \mathcal{Z} \to \mathcal{X}$, where $\mathcal{X}$ is the space of the input data. Such models allow easy sampling, but the computation of the corresponding probability density function is intractable. The goal of these methods is to match $G \# P_Z$ to a target distribution $P_X$ by minimizing $D(P_X, G \# P_Z)$, for some discrepancy $D(\cdot, \cdot)$ between distributions. An overwhelming number of methods have emerged after the introduction of Generative Adversarial Networks [2, 3] and Variational Autoencoders [4] (GANs and VAEs), which have established two distinct paradigms: Adversarial (networks) training and Autoencoders respectively. Adversarial training involves a set of functions $\mathcal{D}$, referred to as *discriminators*, with an objective of the form

$$D(P_X, G \# P_Z) = \max_{d \in \mathcal{D}} \{\mathbb{E}_{x \sim P_X}[a(d(x))] - \mathbb{E}_{x \sim G \# P_Z}[b(d(x))]\}, \tag{1}$$

for some functions $a : \mathbb{R} \to \mathbb{R}$ and $b : \mathbb{R} \to \mathbb{R}$. Autoencoder methods are concerned with finding a function $E : \mathcal{X} \to \mathcal{Z}$, referred to as an *encoder*, whose goal is to reverse $G$, and learn a feature space with the objective

$$D(P_X, G \# P_Z) = \min_E \{\mathcal{R}(G, E) + \Omega(E)\}, \tag{2}$$

where $\mathcal{R}(G, E)$ is the *reconstruction* loss and acts to ensure $G$ and $E$ reverse each other and $\Omega(E)$ is a regularization term. Much work on Autoencoder methods has focused upon the choice of $\Omega$.

In practice, the two methods demonstrate contrasting abilities in their strengths and limitations, which have resulted in differing directions of progress. Indeed, there is a lack of theoretical understanding of how these frameworks are parametrized and it is not clear whether the methods are fundamentally different. For example, Adversarial training based methods have empirically demonstrated high performance when it comes to producing realistic looking samples from $P_X$. However, GANs often have problems in convergence and stability of training [5]. Autoencoders, on the other hand, deal with a more well behaved objective and learn an encoder in the process, making them useful for feature representation. However in practice, Autoencoder based methods have reported shortfalls, such as producing blurry samples for image based datasets [6]. This has motivated researchers to adapt Autoencoder models by borrowing elements from Adversarial networks in the hopes of GAN level performance whilst learning an encoder. Examples include replacing $\Omega$ with Adversarial objectives [7, 8] or replacing the reconstruction loss with an adversarial objective [9, 10]. Recently, the Wasserstein Autoencoder (WAE) [6] has been shown to subsume these two methods with an Adversarial based $\Omega$, and has demonstrated performance similar to that of Adversarial methods.

Understanding the connection between the two paradigms is important for not only the practical purposes outlined above but for the inheritance of theoretical analyses from one another. For example, when it comes to directions of progress, Adversarial training methods now have theoretical guarantees on generalization performance [11], however no such theoretical results have been obtained to date for autoencoders. Indeed, generalization performance is a pressing concern, since both techniques implicitly assume the samples represent the target distribution [12] and eventually leads to memorizing training data.

In this work, we study the two paradigms and in particular focus on the $f$-GANs [3] for Adversarial training and Wasserstein Autoencoders (WAE) for Autoencoders, which generalize the original GAN and VAE models respectively. We prove that the $f$-GAN objective with Lipschitz (with respect to a metric $c$) discriminators is equivalent to the WAE objective with cost $c$. In particular, we show that the WAE objective is an upper bound; schematically we get

$$\boxed{f\text{-GAN} \leq \text{WAE}}$$

and discuss the tightness of this bound. Our result is a generalization of the Kantorovich-Rubinstein duality and thus suggests a primal-dual relationship between Adversarial and Autoencoder methods. Consequently we show, to the best of our knowledge, the first generalization bounds for autoencoders. Furthermore, using this equivalence, we show that the WAE objective is related to key statistical quantities such as the $f$-divergence and Wasserstein distance, which allows us to tap into efficient (regularized) OT solvers.

The main contributions can be summarized as the following:

▷ (Theorem 8) Establishes an equivalence between Adversarial training and Wasserstein Autoencoders, showing conditions under which the $f$-GAN and WAE coincide. This further justifies the similar performance of WAE to GAN based methods. When the conditions are not met, we have an inequality, which allows us to comment on the behavior of the methods.

▷ (Theorem 9, 10 and 14) Show that the WAE objective is related to other statistical quantities such as $f$-divergence and Wasserstein distance.

▷ (Theorem 13) Provide generalization bounds for WAE. In particular, this focuses on the empirical variant of the WAE objective, which allows the use of Optimal Transport (OT) solvers as they are concerned with discrete distributions. This allows one to employ efficient (regularized) OT solvers for the estimation of WAE, $f$-GANs and the generalization bounds.

## 2  Preliminaries

### 2.1  Notation

We will use $\mathcal{X}$ to denote the input space (a Polish space), typically taken to be a Euclidean space. We use $\mathcal{Z}$ to denote the latent space, also taken to be Euclidean. We use $\mathbb{N}_*$ to denote the natural numbers without 0: $\mathbb{N} \setminus \{0\}$. We denote by $\mathscr{P}$ the set of probability measures over $\mathcal{X}$, and elements of this set

will be referred to as *distributions*. If $P \in \mathscr{P}(\mathfrak{X})$ happens to be absolutely continuous with respect to the Lebesgue measure then we will use $dP/d\lambda$ to refer to the *density* function (Radon-Nikodym derivative with respect to the Lebesgue measure). For any $T \in \mathscr{F}(\mathfrak{X}, \mathcal{Z})$, for any measure $\mu \in \mathscr{P}(\mathfrak{X})$, the pushforward measure of $\mu$ through $T$ denoted $T\#\mu \in \mathscr{P}(\mathcal{Z})$ is such that $T\#\mu(A) = \mu(T^{-1}(A))$ for any measurable set $A \subset \mathcal{Z}$. The set $\mathscr{F}(\mathfrak{X}, \mathbb{R})$ refers to all measurable functions from $\mathfrak{X}$ into the set $\mathbb{R}$. We will use functions to represent conditional distributions over a space $\mathcal{Z}$ conditioned on elements $\mathfrak{X}$, for example $P \in \mathscr{F}(\mathfrak{X}, \mathscr{P}(\mathcal{Z}))$ so that for any $x \in \mathfrak{X}$, $P(x) = P(\cdot|x) \in \mathscr{P}(\mathcal{Z})$. For any $P \in \mathscr{P}(\mathfrak{X})$, the *support* of $P$ is $\text{supp}(P) = \{x \in \mathfrak{X} : \text{if } x \in N_x \text{ open} \implies P(N_x) > 0\}$. In any metric space $(\mathfrak{X}, c)$, for any set $S \subseteq \mathfrak{X}$, we define the *diameter* of $S$ to be $\text{diam}_c(S) = \sup_{x,x' \in S} c(x, x')$. Given a metric $c$ over $\mathfrak{X}$, for any $f \in \mathscr{F}(\mathfrak{X}, \mathbb{R})$, $\text{Lip}_c(f)$ denotes the Lipschitz constant of $f$ with respect to $c$ and $\mathcal{H}_c = \{g \in \mathscr{F}(\mathfrak{X}, \mathbb{R}) : \text{Lip}_c(g) \leq 1\}$. For some set $S \subseteq \mathbb{R}$, $\mathbf{1}_S$ corresponds to the convex *indicator function*, ie. $\mathbf{1}_S(x) = 0$ if $x \in S$ and $\mathbf{1}_S(x) = \infty$ otherwise. For any $x \in \mathfrak{X}$, $\delta_x : \mathfrak{X} \to \{0, 1\}$ corresponds to the *characteristic function*, with $\delta_x(0) = 1$ if $x = 0$ and $\delta_x(0) = 0$ if $x \neq 0$.

## 2.2 Background

### 2.2.1 Probability Discrepancies

Probability discrepancies are central to the objective of finding the best fitting model. We introduce some key discrepancies and their notation, which will appear later.

**Definition 1 ($f$-Divergence)** *For a convex function $f : \mathbb{R} \to (-\infty, \infty]$ with $f(1) = 0$, for any $P, Q \in \mathscr{P}(\mathfrak{X})$ with $P$ absolutely continuous with respect to $Q$, the $f$-Divergence between $P$ and $Q$ is*

$$D_f(P, Q) := \int_{\mathfrak{X}} f\left(\frac{dP}{dQ}\right) dQ,$$

*with $D_f(P, Q) = \infty$ if $P$ is note absolutely continuous with respect to $Q$.*

An example of a method to compute the $f$-divergence is to first compute $dP/dQ$ and estimate the integral empirically using samples from $Q$.

**Definition 2 (Integral Probability Metric)** *For a fixed function class $\mathcal{F} \subseteq \mathscr{F}(\mathfrak{X}, \mathbb{R})$, the Integral Probability Metric (IPM) based on $\mathcal{F}$ between $P, Q \in \mathscr{P}(\mathfrak{X})$ is defined as*

$$\text{IPM}_{\mathcal{F}}(P, Q) := \sup_{f \in \mathcal{F}} \left\{ \int_{\mathfrak{X}} f(x) dP(x) - \int_{\mathfrak{X}} f(x) dQ(x) \right\}.$$

If we have that $-\mathcal{F} = \mathcal{F}$ then $\text{IPM}_{\mathcal{F}}$ forms a metric over $\mathscr{P}(\mathfrak{X})$ [13]. A particular IPM we will make use of is Total Variation (TV): $\text{TV}(P, Q) = \text{IPM}_{\mathcal{V}}(P, Q)$ where $\mathcal{V} = \{h \in \mathscr{F}(\mathfrak{X}, \mathbb{R}) : |h| \leq 1\}$. We also note that when $f(x) = |x - 1|$ then $\text{TV} = D_f$ and thus TV is both an IPM and an $f$-divergence.

**Definition 3** *For any $P, Q \in \mathscr{P}(\mathfrak{X})$, define the* set of couplings *between $P$ and $Q$ to be*

$$\Pi(P, Q) = \left\{ \pi \in \mathscr{P}(\mathfrak{X} \times \mathfrak{X}) : \int_{\mathfrak{X}} \pi(x, y) dx = P, \int_{\mathfrak{X}} \pi(x, y) dy = Q \right\}.$$

*For a cost $c : \mathfrak{X} \times \mathfrak{X} \to \mathbb{R}_+$, the* Wasserstein distance *between $P$ and $Q$ is*

$$W_c(P, Q) := \inf_{\pi \in \Pi(P, Q)} \left\{ \int_{\mathfrak{X} \times \mathfrak{X}} c(x, y) d\pi(x, y) \right\}.$$

The Wasserstein distance can be regarded as an infinite linear program and thus admits a dual form, and in the case of $c$ being a metric, belongs to the class of IPMs. We summarize this fact the following lemma [14].

**Lemma 4 (Wasserstein Duality)** *Let $(\mathfrak{X}, c)$ be a metric space, and suppose $\mathcal{H}_c$ is the set of all 1-Lipschitz functions with respect to $c$. Then for any $P, Q \in \mathscr{P}(\mathfrak{X})$, we have*

$$W_c(P, Q) = \sup_{h \in \mathcal{H}_c} \left\{ \int_{\mathfrak{X}} h(x) dP(x) - \int_{\mathfrak{X}} h(x) dQ(x) \right\}$$
$$= \text{IPM}_{\mathcal{H}_c}(P, Q).$$

## 2.3 Generative Models

In both GAN and VAE models, we have a latent space $\mathcal{Z}$ (typically taken to be $\mathbb{R}^d$, with $d$ being small) and a prior distribution $P_Z \in \mathscr{P}(\mathcal{Z})$ (e.g. unit variance Gaussian). We have a function referred to as the generator $G : \mathcal{Z} \to \mathcal{X}$, which induces the *generated* distribution, denoted by $P_G \in \mathscr{P}(\mathcal{X})$, as the pushforward of $P_Z$ through $G$: $P_G = G\#P_Z$. The true data distribution will be referred to as $P_X \in \mathscr{P}(\mathcal{X})$. The common goal between the two methods is to find a generator $G$ such that the samples generated by pushing forward $P_Z$ through $G$ ($G\#P_Z$) are close to the true data distribution ($P_X$). More formally, one can cast this as an optimization problem by finding the best $G$ such that $D(P_G, P_X)$ is minimized, where $D(\cdot, \cdot)$ is some discrepancy between distributions. Both methods (as we outline below) utilize their own discrepancies between $P_X$ and $P_G$, which offer their own benefits and weaknesses.

### 2.3.1 Wasserstein Autoencoder

Let $E : \mathcal{X} \to \mathscr{P}(\mathcal{Z})$ denote a probabilistic *encoder* [1], which maps each point $x$ to a conditional distribution $E(x) \in \mathscr{P}(\mathcal{Z})$, denoted as the *posterior* distribution. The pushforward of $P_X$ through $E$: $E\#P_X$, will be referred to as the *aggregated posterior*.

**Definition 5 (Wasserstein Autoencoder [6])** *Let* $c : \mathcal{X} \times \mathcal{X} \to \mathbb{R}_{\geq 0}$, $\lambda > 0$ *and* $\Omega : \mathscr{P}(\mathcal{Z}) \times \mathscr{P}(\mathcal{Z}) \to \mathbb{R}_{\geq 0}$ *with* $\Omega(P, P) = 0$ *for all* $P \in \mathscr{P}(\mathcal{Z})$. *The Wasserstein Autoencoder objective is*

$$\mathrm{WAE}_{c,\lambda \cdot \Omega}(P_X, G) = \inf_{E \in \mathscr{F}(\mathcal{X}, \mathscr{P}(\mathcal{Z}))} \left\{ \int_{\mathcal{X}} \mathbb{E}_{z \sim E(x)}[c(x, G(z))] dP_X(x) + \lambda \cdot \Omega(E\#P_X, P_Z) \right\}.$$

We remark that there are various choices of $c$ and $\lambda \cdot \Omega$. [6] select these by tuning $\lambda$ and selecting different measures of discrepancies between probability distortions for $\Omega$.

### 2.3.2 $f$-Generative Adversarial Network

Let $d : \mathcal{X} \to \mathbb{R}$ denote a *discriminator* function.

**Definition 6 ($f$-GAN [3])** *Let* $f : \mathbb{R} \to (-\infty, \infty]$ *denote a convex function with property* $f(1) = 0$ *and* $\mathcal{D} \subset \mathscr{F}(\mathcal{X}, \mathbb{R})$ *a set of discriminators. The $f$-GAN model minimizes the following objective for a generator* $G : \mathcal{Z} \to \mathcal{X}$

$$\mathrm{GAN}_f(P_X, G; \mathcal{D}) := \sup_{d \in \mathcal{D}} \{ \mathbb{E}_{x \sim P_X}[d(x)] - \mathbb{E}_{z \sim P_Z}[f^*(d(G(z)))] \}, \tag{3}$$

*where* $f^\star(x) = \sup_y \{x \cdot y - f(y)\}$ *is the convex conjugate of $f$.*

There are two knobs in this method, namely $\mathcal{D}$, the set of discriminators, and the convex function $f$. The objective in (3) is a variational approximation to $D_f$ [3]; if $\mathcal{D} = \mathscr{F}(\mathcal{X}, \mathbb{R})$, then $\mathrm{GAN}_f(P_X, G; \mathcal{D}) = D_f(P_X, P_G)$ [15]. In the case of $f(x) = x \log(x) - (x+1) \log(x+1) + 2 \log 2$, we recover the original GAN [2].

## 3 Related Work

Current attempts at building a taxonomy for generative models have largely been within each paradigm or the proposal of hybrid methods that borrow elements from the two. We first review major and relevant advances in each paradigm, and then move on to discuss results that are close to the technical contributions of our work.

The line of Autoencoders begin with $\Omega = 0$, which is the original autoencoder concerned only with reconstruction loss. VAE then introduced a non-zero $\Omega$, along with implementing Gaussian encoders [4]. This was then replaced by an adversarial objective [7], which is sample based and consequently allows arbitrary encoders. In the spirit of unification, Adversarial Autoencoders (AAE) [8] proposed $\Omega$ to be a discrepancy between the pushforward of the target distribution through the encoder ($E\#P_X$)

and the prior distribution ($P_Z$) in the latent space, which was then showed to be equivalent to the VAE $\Omega$ minus a mutual information term [16]. Independently, InfoVAE [17] proposed a similar objective, which was subsequently shown to be equivalent to adding mutual information. [6] reparametrized the Wasserstein distance into an Autoencoder objective (WAE) where the $\Omega$ term generalizes AAE, and has reported performance comparable to that of Adversarial methods. Other attempts also include adjusting the reconstruction loss to be adversarial as well [9, 10]. Another work that focuses on WAE is the Sinkhorn Autoencoders (SAE) [18], which select $\Omega$ to be the Wasserstein distance and show that the overall objective is an upper bound to the Wasserstein distance between $P_X$ and $P_G$.

[19] discussed the two paradigms and their unification by interpretting GANs from the perspective of variational inference, which allowed a connection to VAE, resulting in a GAN implemented with importance weighting techniques. While this approach is the closest to our work in forming a link, their results apply to standard VAE (and not other AE methods such as WAE) and cannot be extended to all $f$-GANs. [20] introduced the notion of an Adversarial divergence, which subsumed mainstream adversarial based methods. This also led to the formal understanding of how the selected discriminator set $\mathcal{D}$ affects the final generator $G$ learned. However, this approach is silent with regard to Autoencoder based methods. [11] established the tradeoff between the Rademacher complexity of the discriminator class $\mathcal{D}$ and generalization performance of $G$, with no results present for Autoencoders. These theoretical advances in Adversarial training methods are inherited by Autoencoders as a consequence of the equivalence presented in our work.

One key point in the proof of our equivalence is the use of a result that decomposes the GAN objective into an $f$-divergence and an IPM for a restricted class of discriminators (which we used for Lipschitz functions). This decomposition is used in [21] and applied to linear $f$-GANs, showing that the adversarial training objective decomposes into a mixture of maximum likelihood and moment matching. [22] used this decomposition with Lipschitz discriminators like our work, however does not make any extension or further progress to establish the link to WAE. Indeed, GANs with Lipschitz discriminators have been independently studied in [23], which suggest that one should enforce Lipschitz constraints to provide useful gradients.

# 4   $f$-Wasserstein Autoencoders

We define a new objective, that will help us in the proof of the main theorems of this paper.

**Definition 7 ($f$-Wasserstein Autoencoder)** *Let $c : \mathcal{X} \times \mathcal{X} \to \mathbb{R}$, $\lambda > 0$, $f : \mathbb{R} \to (-\infty, \infty]$ be a convex function (with $f(1) = 0$), the $f$-Wasserstein Autoencoder ($f$-WAE) objective is*

$$\overline{W}_{c, \lambda \cdot f}(P_X, G) = \inf_{E \in \mathscr{F}(\mathcal{X}, \mathscr{P}(\mathcal{Z}))} \{W_c(P_X, (G \circ E) \# P_X) + \lambda D_f(E \# P_X, P_Z)\} \qquad (4)$$

In the proof of the main result, we will show that the $f$-WAE objective is indeed the same as the WAE objective when using the same cost $c$ and selecting the regularizer to be $\lambda \cdot \Omega = D_{\lambda f} = \lambda D_f$. The only difference between this and the standard WAE is the use of $W_c(P_X, (G \circ E) \# P_X)$ as the reconstruction loss instead of the standard cost which is an upper bound (Lemma 18), and the regularizer is chosen to be $\lambda \cdot \Omega = D_{\lambda f} = \lambda D_f$. We now present the main theorem that captures the relationship between $f$-GAN and WAE.

**Theorem 8 ($f$-GAN and WAE equivalence)** *Suppose $(\mathcal{X}, c)$ is a metric space and let $\mathcal{H}_c$ denote the set of all functions from $\mathcal{X} \to \mathbb{R}$ that are 1-Lipschitz (with respect to c). Let $f : \mathbb{R} \to (-\infty, \infty]$ be a convex function with $f(1) = 0$. Then for all $\lambda > 0$,*

$$\text{GAN}_{\lambda f}(P_X, G; \mathcal{H}_c) \leq \text{WAE}_{c, \lambda \cdot D_f}(P_X, G), \qquad (5)$$

*with equality if $G$ is invertible.*

**Proof (This is a sketch, see Section A.1 for full proof)**. The proof begins by proving certain properties of $\mathcal{H}_c$ (Lemma 16), allowing us to use the dual form of restricted GANs (Theorem 15),

$$\text{GAN}_f(P_X, G; \mathcal{H}_c) = \inf_{P' \in \mathscr{P}(\mathcal{X})} \left\{ D_f(P', P_G) + \sup_{h \in \mathcal{H}_c} \{\mathbb{E}_{P_X}[h] - \mathbb{E}_{P'}[h]\} \right\}$$
$$= \inf_{P' \in \mathscr{P}(\mathcal{X})} \{D_f(P', P_G) + W_c(P', P_X)\}. \qquad (6)$$

The key is to reparametrize (6) by optimizing over couplings. By rewriting $P' = (G \circ E) \# P_X$ for some $E \in \mathscr{F}(\mathfrak{X}, \mathscr{P}(\mathfrak{Z}))$ and rewriting (6) as an optimization over $E$ (Lemma 20), we obtain

$$\inf_{P' \in \mathscr{P}(\mathfrak{X})} \{D_f(P', P_G) + W_c(P', P_X)\}$$
$$= \inf_{E \in \mathscr{F}(\mathfrak{X}, \mathscr{P}(\mathfrak{Z}))} \{D_f((G \circ E) \# P_X, P_G) + W_c((G \circ E) \# P_X, P_X)\} \tag{7}$$

We then have

$$D_f((G \circ E) \# P_X, P_G) = D_f(G \# (E \# P_X), G \# P_Z) \overset{(*)}{\leq} D_f(E \# P_X, P_Z),$$

with equality in $(*)$ if $G$ is invertible (Lemma 17). A weaker condition is required if $f$ is differentiable, namely if $G$ is invertible with respect to $f' \circ d(E \# P_X)/dP_Z$ in the sense that

$$G(z) = G(z') \implies f' \circ (d(E \# P_X)/dP_Z)(z) = f' \circ (d(E \# P_X)/dP_Z)(z'), \tag{8}$$

noting that an invertible $G$ trivially satisfies this requirement. Letting $f \leftarrow \lambda f$, we have $D_f(\cdot, \cdot) \leftarrow \lambda D_f(\cdot, \cdot)$, and so from Equation 7, we have

$$\text{GAN}_{\lambda f}(P_X, G; \mathcal{H}_c) \overset{(*)}{\leq} \inf_{E \in \mathscr{F}(\mathfrak{X}, \mathscr{P}(\mathfrak{Z}))} \{\lambda D_f(E \# P_X, P_Z) + W_c((G \circ E) \# P_X, P_X)\}$$
$$= \overline{W}_{c, \lambda \cdot f}(P_X, G)$$
$$\leq \inf_{E \in \mathscr{F}(\mathfrak{X}, \mathscr{P}(\mathfrak{Z}))} \left\{\lambda D_f(E \# P_X, P_Z) + \int_{\mathfrak{X}} \mathbb{E}_{z \sim E(x)}[c(x, G(z))] dP_X(x)\right\}$$
$$= \text{WAE}_{c, \lambda \cdot D_f}(P_X, G),$$

where the final inequality follows from the fact that $W_c(P, Q) \leq \int_{\mathfrak{X}} \mathbb{E}_{z \sim E(x)}[c(x, G(z))] dP_X(x)$ (Lemma 18). Using the fact that $\overline{W} \geq \text{WAE}$ (Lemma 19) completes the proof. ∎

When $G$ is invertible, we remark that $P_G$ can still be expressive and capable of modelling complex distributions in WAE and GAN models. For example, if $G$ is implemented with feedforward neural networks, and $G$ is invertible then $P_G$ can model *deformed* exponential families [24], which encompasses a large class appearing in statistical physics and information geometry [25, 26]. There exists many invertible activation functions under which $G$ will be invertible. Furthermore, in the proof of the Theorem it is clear that $\overline{W}$ and WAE are the same objective (from Lemma 18 and Lemma 19). When using $f = \mathbf{1}_{\{1\}}$ ($f(x) = 0$ if $x = 1$ and $f(x) = \infty$ otherwise), and noting that $f^\star(x) = x$, meaning that Theorem 8 (with $\lambda = 1$) reduces to

$$\sup_{h \in \mathcal{H}_c} \{\mathbb{E}_{x \sim P_X}[h(x)] - \mathbb{E}_{x \sim P_G}[h(x)]\} = \text{GAN}_f(P_X, G; \mathcal{H}_c)$$
$$\leq \overline{W}_{c, f}(P_X, P_G)$$
$$= \inf_{E \in \mathscr{F}(\mathfrak{X}, \mathscr{P}(\mathfrak{Z})): E \# P_X = P_Z} \{W_c(P_X, (G \circ E) \# P_X)\}$$
$$= \inf_{E \in \mathscr{F}(\mathfrak{X}, \mathscr{P}(\mathfrak{Z})): E \# P_X = P_Z} \{W_c(P_X, G \# P_Z)\}$$
$$= W_c(P_X, P_G),$$

which is the standard primal-dual relation between Wasserstein distances as in Lemma 4. Hence, Theorem 8 can be viewed as a generalization of this primal-dual relationship, where Autoencoder and Adversarial objectives represent primal and dual forms respectively.

We note that the left hand of Equation (5) does not explicitly engage the prior space $Z$ as much as the right hand side in the sense that one can set $Z = \mathfrak{X}$, $G = \text{Id}$ (which is invertible) and $P_Z = P_G$ and indeed results in the exact same $f$-GAN objective since $G \# P_Z = \text{Id} \# P_G = P_G$, yet the equivalent $f$-WAE objective (from Theorem 8) will be different. This makes the Theorem versatile in reparametrizations, which we exploit in the proof of Theorem 10. We now consider weighting the reconstruction along with the regularization term in $\overline{W}$ (which is equivalent to weighting WAE), which simply amounts to re-weighting the cost since for any $\gamma > 0$,

$$\overline{W}_{\gamma \cdot c, \lambda \cdot f}(P_X, G) = \inf_{E \in \mathscr{F}(\mathfrak{X}, \mathscr{P}(\mathfrak{Z}))} \{\gamma W_c((G \circ E) \# P_X, P_X) + \lambda D_f(E \# P_X, P_Z)\}.$$

The idea of weighting the regularization term by $\lambda$ was introduced by [27] and furthermore studied empirically, showing that the choice of $\lambda$ influences learning disentanglement in the latent space. [28]. We show that if $\lambda = 1$ and $\gamma$ is larger than some $\gamma^*$ then $\overline{W}$ will become an $f$-divergence (Theorem 9). On the other hand if we fix $\gamma = 1$ and take $\lambda$ is larger than some $\lambda^*$, then $\overline{W}$ becomes the Wasserstein distance and in particular, equality holds in (5) (Theorem 10). We show explicitly how high $\gamma$ and $\lambda$ need to be for such equalities to occur. This is surprising since $f$-divergence and Wasserstein distance are quite different distortions.

We begin with the $f$-divergence case. Consider $f : \mathbb{R} \to (-\infty, \infty]$ convex, differentiable and $f(1) = 0$ and assume that $P_X$ is absolutely continuous with respect to $P_G$, so that $D_f(P_X, P_G) < \infty$.

**Theorem 9** *Set $c(x, y) = \delta_{x-y}$ and let $f : \mathbb{R} \to (-\infty, \infty]$ be a convex function (with $f(1) = 0$) and differentiable. Let $\gamma^* = \sup_{x,x' \in \mathcal{X}} \left| f'\left(\frac{dP_X}{dP_G}\right) - f'(\frac{dP_X}{dP_G})(x') \right|$ and suppose $P_G$ is absolutely continuous with respect to $P_X$ and that $G$ is invertible, then we have for all $\gamma \geq \gamma^*$*

$$\overline{W}_{\gamma \cdot c, f}(P_X, G) = D_f(P_X, P_G).$$

(Proof in Appendix, Section A.3). It is important to note that $W_c(P_X, P_G) = \text{TV}(P_X, P_G)$ when $c(x, y) = \delta_{x-y}$ and so Theorem 9 tells us that the objective with a weighted total variation reconstruction loss with a $f$-divergence prior regularization amounts to the $f$-divergence. It was shown that in [24] that when $G$ is an invertible feedforward neural network then $D_f(P_X, P_G)$ is a *Bregman* divergence (a well regarded quantity in information geometry) between the parametrizations of the network for a fixed choice of activation function of $G$, which depends on $f$. Hence, a practioner should design $G$ with such activation function when using $f$-WAE under the above setting ($c(x, y) = \delta_{x-y}$ and $\gamma = \gamma^*$) with $G$ being invertible, so that the information theoretic divergence ($D_f$) between the distributions becomes an information geometric divergence involving the network parameters.

We now show that if $\lambda$ is selected higher than $\lambda^* := \sup_{P' \in \mathscr{P}(\mathcal{X})} (W_c(P', P_G)/D_f(P', P_G))$, then $\overline{W}$ becomes $W_c$ and furthermore we have equality between $f$-GAN, $f$-WAE and WAE.

**Theorem 10** *Let $c : \mathcal{X} \times \mathcal{X} \to \mathbb{R}$ be a metric. For any $f : \mathbb{R} \to (-\infty, \infty]$ convex function (with $f(1) = 0$), we have for all $\lambda \geq \lambda^*$*

$$\text{GAN}_{\lambda f}(P_X, G; \mathcal{H}_c) = \overline{W}_{c, \lambda \cdot f}(P_X, G) = \text{WAE}_{c, \lambda \cdot D_f}(P_X, G) = W_c(P_X, P_G).$$

(Proof in Appendix, Section A.4). Note that Theorem 10 holds for any $f$ (satisfying properties of the Theorem) and so one can estimate the Wasserstein distance using any $f$ as long as $\lambda$ is scaled to $\lambda^*$. In order to understand how high $\lambda^*$ can be, there are two extremes in which the supremum may be unbounded. The first case is when $P'$ taken far from $P_G$ so that $W_c(P', P_G)$ increases, however one should note that in the case when $\Delta = \max_{x,x' \in \mathcal{X}} c(x, x') < \infty$ then $W_c \in [0, \Delta]$ and so $W_c$ will be finite whereas $D_f(P', P_G)$ can possibly diverge to $\infty$, making $\lambda^* \to 0$. The other case is when $P'$ is made close to $P_G$, in which case $\frac{1}{D_f(P', P_G)} \to \infty$ however $W_c(P', P_G) \to 0$ so the quantity $\lambda^*$ can still be small in this case, depending on the rate of decrease between $W_c$ and $D_f$. Now suppose that $f(x) = |x - 1|$ and $c(x, y) = \delta_{x-y}$, in which case $D_f = W_c$ and thus $\lambda^* = 1$. In this case, Theorem 10 reduces to the standard result [29] regarding the equivalence between Wasserstein distance and $f$-divergence intersecting at the variational divergence under these conditions.

## 5 Generalization bounds

We present generalization bounds using machinery developed in [30] with the following definitions.

**Definition 11 (Covering Numbers)** *For a set $S \subseteq \mathcal{X}$, we denote $N_\eta(S)$ to be the $\eta$-covering number of $S$, which is the smallest $m \in \mathbb{N}_*$ such that there exists closed balls $B_1, \ldots, B_m$ of radius $\eta$ with $S \subseteq \bigcup_{i=1}^m B_i$. For any $P \in \mathscr{P}(\mathcal{X})$, the $(\eta, \tau)$-dimension is $d_\eta(P, \tau) := \frac{\log N_\eta(P, \tau)}{-\log \eta}$, where $N_\eta(P, \tau) := \inf \{ N_\eta(S) : P(S) \geq 1 - \tau \}$.*

**Definition 12 (1-Upper Wasserstein Dimension)** *The* 1-Upper Wasserstein dimension *of any* $P \in \mathscr{P}(\mathfrak{X})$ *is* $d^*(P) := \inf \left\{ s \in (2, \infty) : \limsup_{\eta \to 0} d_\eta(P, \eta^{\frac{s}{s-2}}) \leq s \right\}$.

We make an assumption of $P_X$ and $P_G$ having bounded support to achieve the following bounds. For any $P \in \mathscr{P}(\mathfrak{X})$ in a metric space $(\mathfrak{X}, c)$, we use define $\Delta_{P,c} = \text{diam}_c(\text{supp}(P))$.

**Theorem 13** *Let* $(\mathfrak{X}, c)$ *be a metric space and suppose* $\Delta := \max \{ \Delta_{c,P_X}, \Delta_{c,P_G} \} < \infty$. *For any* $n \in \mathbb{N}_*$, *let* $\hat{P}_X$ *and* $\hat{P}_G$ *denote the empirical distribution with* $n$ *samples drawn i.i.d from* $P_X$ *and* $P_G$ *respectively. Let* $s_X > d^*(P_X)$ *and* $s_G > d^*(P_G)$. *For all* $f : \mathbb{R} \to (-\infty, \infty]$ *convex functions,* $f(1) = 0$, $\lambda > 0$ *and* $\delta \in (0, 1)$, *then with probability at least* $1 - \delta$, *we have*

$$\text{GAN}_{\lambda f}(P_X, G; \mathcal{H}_c) \leq \overline{W}_{c, \lambda \cdot f}(\hat{P}_X, P_G) + O\left( n^{-1/s_X} + \Delta \sqrt{\frac{1}{n} \ln\left(\frac{1}{\delta}\right)} \right), \tag{9}$$

*and if* $f(x) = |x - 1|$ *is chosen then*

$$\text{GAN}_{\lambda f}(P_X, G; \mathcal{H}_c) \leq \overline{W}_{c, \lambda \cdot f}(\hat{P}_X, \hat{P}_G) + O\left( n^{-1/s_X} + n^{-1/s_G} + \Delta \sqrt{\frac{1}{n} \ln\left(\frac{1}{\delta}\right)} \right). \tag{10}$$

(Proof in Appendix, Section A.2). First note that there is no requirement on $G$ to be invertible and no restriction on $\lambda$. Second, there are the quantities $s_X$, $s_G$ and $\Delta$ that are influenced by the distributions $P_X$ and $P_G$. It is interesting to note that $d^*$ is related to fractal dimensions [31] and thus relates the convergence of GANs to statistical geometry. If $G$ is invertible in the above then the left hand side of both bounds becomes $\overline{W}_{c, \lambda \cdot f}(P_X, G)$ by Theorem 8. In general, $\hat{P}_X$ and $\hat{P}_G$ will not share the same support, in which case $D_f(\hat{P}_X, \hat{P}_G) = \infty$ – This would lead one to suspect the same from $\overline{W}_{c, \lambda \cdot f}(\hat{P}_X, \hat{P}_G)$, however this is not the case since

$$\overline{W}_{c, \lambda \cdot f}(\hat{P}_X, \hat{P}_G) \leq \inf_{E \in \mathscr{F}(\mathfrak{X}, \mathscr{P}(\mathfrak{X}))} \left\{ W_c((G \circ E) \# P_X, P_X) + \lambda D_f(E \# \hat{P}_X, \hat{P}_Z) \right\},$$

and so $E \in \mathscr{F}(\mathfrak{X}, \mathscr{P}(\mathfrak{Z}))$ would be selected such that $E \# \hat{P}_X$ shares the support of $\hat{P}_Z$, resulting in a bounded value. We now show the relationship between $\overline{W}$ and $W_c$.

**Theorem 14** *For any* $c : \mathfrak{X} \times \mathfrak{X} \to \mathbb{R}$, $\lambda > 0$ *and* $f : \mathbb{R} \to (-\infty, \infty]$ *convex function (with* $f(1) = 0$*) we have* $\overline{W}_{c, \lambda \cdot f}(P_X, G) \leq W_c(P_X, P_G)$

(Proof in Appendix, Section A.5). This suggests that in order to minimize $\overline{W}$, one can minimize $W_c$. Indeed, majority of the solvers are concerned with discrete distributions, which is exactly what is present on the right hand side of the generalization bounds: $\overline{W}_{c, \lambda \cdot f}(\hat{P}_X, \hat{P}_G)$

## 6 Discussion and Conclusion

This work is the first to prove a generalized primal-dual betweenship between GANs and Autoencoders. Our result elucidated the close performance between WAE and $f$-GANs. Furthermore, we explored the effect of weighting the reconstruction and regularization on the WAE objective, showing relationships to both $f$-divergences and Wasserstein metrics along with the impact on the duality relationship. This equivalence allowed us to prove generalization results, which to the best of our knowledge, are the first bounds given for Autoencoder models. The results imply that we can employ efficient (regularized) OT solvers to approximate upper bounds on the generalization bounds, which involve discrete distributions and thus are natural for such solvers.

The consequences of unifying two paradigms are plentiful, generalization bounds being an example. One line of extending and continuing the presented work is to explore the use of a general cost $c$ (as opposed to a metric), invoking the generalized Wasserstein dual in the goal of forming a generalized GAN. Our paper provides a basis to unify Adversarial Networks and Autoencoders through a primal-dual relationship, and opens doors for the further unification of related models.

**Acknowledgments**

We would like to acknowledge anonymous reviewers and the Australian Research Council of Data61.

## Footnotes

[1] We remark that this is not standard notation in the VAE and Variational Inference literature.

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
