[Supplementary Material]

# A  Appendix

## A.1  Proof of Theorem 8

In order to prove the theorem, we make use of the dual form of the restricted variational form of an $f$-divergence:

**Theorem 15 ([21], Theorem 3)** *Let $f : \mathbb{R} \to (-\infty, \infty]$ denote a convex function with property $f(1) = 0$ and suppose $H$ is a convex subset of $\mathscr{F}(\mathfrak{X}, \mathbb{R})$ with the property that for any $h \in H$ and $b \in \mathbb{R}$, we have $h + b \in H$. Then for any $P, Q \in \mathscr{P}(\mathfrak{X})$ we have*

$$\sup_{h \in H} \{\mathbb{E}_{x \sim P}[h(x)] - \mathbb{E}_{x \sim Q}[f^*(h(x))]\} = \inf_{P' \in \mathscr{P}(\mathfrak{X})} \left\{ D_f(P', Q) + \sup_{h \in H} \{\mathbb{E}_P[h(x)] - E_{P'}[h(x)]\} \right\}$$

The goal is now to set $H = \mathcal{H}_c$ however there are some conditions of the above that we require

**Lemma 16** *If $c$ is a metric then $\mathcal{H}_c$ is convex and closed under addition.*

**Proof**  Let $f \in \mathcal{H}_c$ and consider define $h = f + b$ for some $b \in \mathbb{R}$, we then have

$$\begin{aligned}
|h(x) - h(y)| &= |f(x) + b - f(y) - b| \\
&= |f(x) - f(y)| \\
&\leq c(x, y)
\end{aligned}$$

Consider some $\lambda \in [0, 1]$ and set $h(x) = \lambda \cdot f(x) + (1 - \lambda) \cdot g(x)$ for some $f, g \in \mathcal{H}_c$. We then have

$$\begin{aligned}
|h(x) - h(y)| &= |\lambda \cdot f(x) + (1 - \lambda) \cdot g(x) - \lambda \cdot f(y) - (1 - \lambda) \cdot g(y)| \\
&= |\lambda \cdot (f(x) - f(y)) + (1 - \lambda) \cdot (g(x) - g(y))| \\
&\leq \lambda \cdot |f(x) - f(y)| + (1 - \lambda) \cdot |g(x) - g(y)| \\
&\leq \lambda \cdot c(x, y) + (1 - \lambda) \cdot c(x, y) \\
&= c(x, y)
\end{aligned}$$

for all $x, y \in \mathfrak{X}$. ∎

We require a lemma regarding the decomposibility of $G$ for $f$-divergences.

**Lemma 17** *Let $G : \mathcal{Z} \to \mathfrak{X}$ and let $P, Q$ be two distributions over $\mathcal{Z}$. We have that*

$$D_f(G\#P, G\#Q) \leq D_f(P, Q),$$

*with equality if $G$ is invertible. Furthermore, if $f$ is differentiable then we have equality for a weaker condition: for any $z, z' \in \mathcal{Z}, G(z) = G(z') \implies f'(\frac{dP}{dQ}(z)) = f'(\frac{dP}{dQ}(z')).$*

**Proof**  By writing the variational form from [15] (Lemma 1), we have

$$\begin{aligned}
D_f(G\#P, G\#Q) &= \sup_{h \in \mathscr{F}(\mathfrak{X}, \mathbb{R})} \{\mathbb{E}_{x \sim G\#P}[h(x)] - \mathbb{E}_{x \sim G\#Q}[f^*(h(x))]\} \\
&= \sup_{h \in \mathscr{F}(\mathfrak{X}, \mathbb{R})} \{\mathbb{E}_{z \sim P}[h(G(z))] - \mathbb{E}_{z \sim Q}[f^*(h(G(z)))]\} \\
&= \sup_{h \in \mathscr{F}(\mathfrak{X}, \mathbb{R}) \circ G} \{\mathbb{E}_{z \sim P}[h(z)] - \mathbb{E}_{z \sim Q}[f^*(h(z))]\} \\
&\leq \sup_{h \in \mathscr{F}(\mathcal{Z}, \mathbb{R})} \{\mathbb{E}_{z \sim P}[h(z)] - \mathbb{E}_{z \sim Q}[f^*(h(z))]\} \\
&= D_f(P, Q),
\end{aligned}$$

where we used the fact that $\mathscr{F}(\mathfrak{X}, \mathbb{R}) \circ G \subseteq \mathscr{F}(\mathcal{Z}, \mathbb{R})$. If $G$ is invertible then we applying the above with $G \leftarrow G^{-1}$, $P \leftarrow G\#P$ and $Q \leftarrow G\#Q$, we have

$$D_f(G^{-1}\#(G\#P), G^{-1}\#(G\#Q)) \leq D_f(G\#P, G\#Q),$$

which is just the reverse direction $D_f(P, Q) \leq D_f(G\#P, G\#Q)$, and so equality holds. Suppose now that $f$ is differentiable then note that inequality holds when $f'(dP/dQ) \in \mathscr{F}(\mathcal{X}, \mathbb{R}) \circ G$ (See proof of Lemma 1 in [15]), which is equivalent to asking if there exists a function $\varphi_f \in \mathscr{F}(\mathcal{X}, \mathbb{R})$ such that

$$\varphi_f \circ G = f' \left( \frac{dP}{dQ} \right).$$

For any $z \in \mathcal{Z}$, we can construct $\varphi_f$ to map $G(z)$ to $f' \left( \frac{dP}{dQ} \right)(z)$ and due to the condition in the lemma, we can guarantee $\varphi_f$ will indeed be a function and thus exists. ∎

We need a Lemma that will allow us to upper bound the Wasserstein distance.

**Lemma 18** *For any $E \in \mathscr{F}(\mathcal{X}, \mathscr{P}(\mathcal{Z}))$, $G \in \mathscr{F}(\mathcal{Z}, \mathcal{X})$ and $c : \mathcal{X} \times \mathcal{X} \to \mathbb{R}$, we have*

$$W_c((G \circ E)\#P_X, P_X) \leq \int_{\mathcal{X}} \mathbb{E}_{z \sim E(x)}[c(x, G(z))]dP_X(x).$$

**Proof** We quote a reparametrization result from [6] Theorem 1 that if $G$ is deterministic then the Wasserstein distance can be reparametrized as

$$W_c(G\#(E\#P_X), P_X) = \inf_{Q \in \mathscr{F}(\mathcal{X}, \mathscr{P}(\mathcal{Z})): Q\#P_X = E\#P_X} \int_{\mathcal{X}} \mathbb{E}_{z \sim Q(x)}[c(x, G(z))]dP_X(x) \quad (11)$$

$$\leq \int_{\mathcal{X}} \mathbb{E}_{z \sim E(x)}[c(x, G(z))]dP_X(x).$$

∎

We also need a Lemma regarding the relationship between $\overline{W}$ and WAE.

**Lemma 19** *Let $f : \mathbb{R} \to (-\infty, \infty]$ be a convex function with $f(1) = 0$, then we have*

$$\overline{W}_{c, \lambda \cdot f}(P_X, G) \leq \text{WAE}_{c, \lambda \cdot D_f}(P_X, G).$$

**Proof** Consider the optimal encoder $E^*$ from the $f$-WAE objective. Let $Q^* = E^*\#P_X$. We then have that

$$\overline{W}_{c, \lambda \cdot f}(P_X, G) = W_c(P_X, G\#Q^*) + \lambda \cdot D_f(Q^*, P_Z).$$

Let $\pi \in \Pi(P_X, E\#Q^*)$ be the optimal coupling under the metric $c$. By the Gluing lemma [14], one can construct a triple $(X, Y, Z)$ where $(X, Y) \sim \pi$, $Z \sim Q^*$ and $Y = G(Z)$ almost surely. Let $\pi'$ be the distribution over $(Y, Z)$ and consider the conditional distribution over $Z$ given $Y$, associated with $E_{\pi'} \in \mathscr{F}(\mathcal{X}, \mathscr{P}(\mathcal{Z}))$. We have $E_{\pi'}\#P_X = Q^*$ and so we have

$$\begin{aligned}
\text{WAE}_{c, \lambda \cdot D_f}(P_X, G) &\leq \int_{\mathcal{X}} \mathbb{E}_{z \sim E_{\pi'}(y)}[c(x, G(z))]dP_X + D_f(E_{\pi'}\#P_X, P_Z) \\
&= \int_{\mathcal{X}} \mathbb{E}_{z \sim E_{\pi'}(y)}[c(x, G(z))]dP_X + D_f(Q^*, P_Z) \\
&= \int_{\mathcal{X} \times \mathcal{X}} [c(x, y)]d\pi'(x, y) + D_f(Q^*, P_Z) \\
&= W_c(P_X, G\#Q^*) + \lambda \cdot D_f(Q^*, P_Z). \\
&= \overline{W}_{c, \lambda \cdot f}(P_X, G).
\end{aligned}$$

∎

Finally, we need a lemma to justify reparametrizations.

**Lemma 20** *If $G : \mathcal{Z} \to \mathcal{X}$ is invertible then for any $P' \in \mathscr{P}(\mathcal{X})$ such that $P' \ll P_G$, then there exists an $E \in \mathscr{F}(\mathcal{X}, \mathscr{P}(\mathcal{Z}))$ such that $P' = G\#E\#P_X$.*

**Proof** From the assumption, we have $\text{Supp}(P') \subseteq \text{Supp}(P_G) \subseteq \text{Im}(G)$, and so by invertibility of $G$, we can set $Q = G^{-1}\#P'$ and construct a conditional distribution $E$ (between marginals $Q$ and $P_X$) to get $Q = E\#P_X$, hence $P' = G\#E\#P_X$. ∎

We are now ready to prove the theorem. Set $H = \mathcal{H}_c$ (the set of 1-Lipschitz functions) and note that $\lambda f$ is a convex function satisfying $\lambda f(1) = 0$ and so substituting $f \leftarrow \lambda f$, we get that $D_{\lambda f}(\cdot, \cdot) = \lambda D_f(\cdot, \cdot)$. Hence, we have

$$
\begin{aligned}
\text{GAN}_{\lambda f}(P_X, G; \mathcal{H}_c) &= \sup_{h \in H_c} \{\mathbb{E}_{x \sim P_X}[h(x)] - \mathbb{E}_{x \sim P_G}[(\lambda f)^\star(h(x))]\} \\
&= \inf_{P' \in \mathscr{P}(\mathfrak{X})} \{\lambda D_f(P', P_G) + W_c(P', P_X)\} \\
&= \inf_{P' \in \mathscr{P}(\mathfrak{X}): P' << P_g} \{\lambda D_f(P', P_G) + W_c(P', P_X)\} \\
&= \inf_{E \in \mathscr{F}(\mathfrak{X}, \mathscr{P}(\mathfrak{Z}))} \{\lambda D_f((G \circ E)\#P_X, G\#P_Z) + W_c((G \circ E)\#P_X, P_X)\}
\end{aligned}
$$
(12)

$$
\begin{aligned}
&\overset{(*)}{\leq} \inf_{E \in \mathscr{F}(\mathfrak{X}, \mathscr{P}(\mathfrak{Z}))} \{\lambda D_f(E\#P_X, P_Z) + W_c((G \circ E)\#P_X, P_X)\} \\
&= \overline{W}_{c, \lambda \cdot f}(P_X, G) \\
&\leq \inf_{E \in \mathscr{F}(\mathfrak{X}, \mathscr{P}(\mathfrak{Z}))} \left\{ \int_{\mathfrak{X}} \mathbb{E}_{z \sim E(x)}[c(x, G(z))]dP_X(x) + \lambda D_f(E\#P_X, P_Z) \right\} \\
&= \text{WAE}_{c, \lambda \cdot D_f}(P_X, G),
\end{aligned}
$$
(13)

where (12) is an equality when $G$ is invertible from Lemma 20 and $(*)$ is $=$ if $G$ satisfies the requirement of Lemma 17. To prove the final inequality, note that if $E^*$ satisfies the condition of the Theorem then

$$
\begin{aligned}
\overline{W}_{c, \lambda \cdot f}(P_X, G) &= W_c((G \circ E^*)\#P_X, P_X) + \lambda D_f(E^*\#P_X, P_Z) \\
&= W_c(G\#(E^*\#P_X), P_X) \\
&= W_c(P_G, P_X).
\end{aligned}
$$
(14)

Next, notice that

$$
\begin{aligned}
&\text{WAE}_{c, \lambda \cdot D_f}(P_X, G) \\
&= \inf_{E \in \mathscr{F}(\mathfrak{X}, \mathscr{P}(\mathfrak{Z}))} \left\{ \int_{\mathfrak{X}} \mathbb{E}_{z \sim E(x)}[c(x, G(z))]dP_X(x) + \lambda D_f(E\#P_X, P_Z) \right\} \\
&\leq \inf_{E \in \mathscr{F}(\mathfrak{X}, \mathscr{P}(\mathfrak{Z})): E\#P_X = P_Z} \left\{ \int_{\mathfrak{X}} \mathbb{E}_{z \sim E(x)}[c(x, G(z))]dP_X(x) + \lambda D_f(E\#P_X, P_Z) \right\} \\
&\leq \inf_{E \in \mathscr{F}(\mathfrak{X}, \mathscr{P}(\mathfrak{Z})): E\#P_X = P_Z} \left\{ \int_{\mathfrak{X}} \mathbb{E}_{z \sim E(x)}[c(x, G(z))]dP_X(x) \right\} \\
&= W_c(P_X, P_G)
\end{aligned}
$$
(15)
$$
= \overline{W}_{c, \lambda \cdot f}(P_X, G),
$$
(16)

where (15) follows from the reparametrized Wasserstein distance from [6] (Theorem 1), which we used in (11) and the final step follows from (14). Combining $\text{WAE}_{c, \lambda \cdot D_f}(P_X, G) \leq \overline{W}_{c, \lambda \cdot f}(P_X, G)$ with $\text{WAE}_{c, \lambda \cdot D_f}(P_X, G) \geq \overline{W}_{c, \lambda \cdot f}(P_X, G)$ (from 13) yields equality and concludes the proof.

## A.2 Proof of Theorem 13

We first prove a lemma that will apply to both cases. Recalling that for any metric space $(\mathfrak{X}, c)$ and $P \in \mathscr{P}(\mathfrak{X})$ we define $\Delta_{P,c} = \text{diam}_c(\text{supp}(P))$.

**Lemma 21** *Let $(\mathfrak{X}, c)$ be a metric space. For any $P \in \mathscr{P}(\mathfrak{X})$, suppose $\Delta_{P,c} < \infty$ and let $\hat{P}$ denote the empirical distribution after drawing $n$ i.i.d samples for some $n \in \mathbb{N}_*$. If $s > d^*(P)$, then we have*

$$
\text{IPM}_{\mathcal{H}_c}(P, \hat{P}) \leq O(n^{-1/s}) + \frac{\Delta_{P,c}}{2}\sqrt{\frac{2}{n}\ln\left(\frac{1}{\delta}\right)}
$$

**Proof** We appeal to McDiarmind's Inequality and use a standard method, as shown in [32], to bound the quantity.

**Theorem 22 (McDiarmind's Inequality)** *Let $X_1, \ldots, X_n$ be $n$ independent random variables and consider a function $\Phi : \mathcal{X}^n \to \mathbb{R}$ such that there exists constants $c_i > 0$ (for $i = 1, \ldots, n$) with*

$$\sup_{x_1, \ldots, x_n, x_i'} |\Phi(x_1, \ldots, x_n) - \Phi(x_1, \ldots, x_{i-1}, x_i', x_{i+1}, \ldots, x_n)| \leq c_i.$$

*Then for any $t > 0$, we have*

$$\Pr\left[\Phi(X_1, \ldots, X_n) - \mathbb{E}\left[\Phi(X_1, \ldots, X_n)\right] \geq t\right] \leq \exp\left(\frac{-2t^2}{\sum_{i=1}^n c_i^2}\right)$$

Let $\mathcal{F} = \mathcal{H}_c$ then let

$$\Phi(S) = \text{IPM}_{\mathcal{H}_c}(P, \hat{P}).$$

Noting that

$$|\Phi(x_1, \ldots, x_n) - \Phi(x_1, \ldots, x_{i-1}, x_i', x_{i+1}, \ldots, x_n)| \leq \frac{1}{n}|f(x_i) - f(x_i')|$$

$$\leq \frac{1}{n} \cdot c(x_i, x_i')$$

$$\leq \frac{\Delta_{P,c}}{n},$$

where the first inequality follows as each $f$ is 1-Lipschitz and the second follows from the fact that each $x, x' \in \text{supp}(P)$. This allows us to set $c_i = \Delta/n$ for all $i = 1, \ldots, n$. Now applying McDiarmind's inequality with $t = \Delta_{P,c}/2\sqrt{\frac{2}{n}\ln\left(\frac{1}{\delta}\right)}$ yields (for a sample $S \sim P^n$)

$$\Pr\left[\Phi(S) - \mathbb{E}\Phi(S) \geq \frac{\Delta_{P,c}}{2}\sqrt{\frac{2}{n}\ln\left(\frac{1}{\delta}\right)}\right] \leq \delta$$

$$\Pr\left[\Phi(S) - \mathbb{E}\Phi(S) \leq \frac{\Delta_{P,c}}{2}\sqrt{\frac{2}{n}\ln\left(\frac{1}{\delta}\right)}\right] \geq 1 - \delta,$$

and thus

$$\Phi(S) \leq \mathbb{E}\Phi(S) + \frac{\Delta_{P,c}}{2}\sqrt{\frac{2}{n}\ln\left(\frac{1}{\delta}\right)}.$$

Noting that $\mathbb{E}\Phi(S) = \mathbb{E}[W_c(P, \hat{P})]$ (from Lemma 4), we appeal to a case of Theorem 1 in [30] where $p = 1$, which tells us that if $s > d^*(P)$ then $\mathbb{E}[W_c(P, \hat{P})] = O(n^{-1/s})$. Since this is the requirement in the lemma, the proof concludes. ∎

We will make use of this lemma for both $P_X$ and $P_G$ and use $\Delta$ for both cases since $\Delta \geq \Delta_{P_X,c}$ and $\Delta \geq \Delta_{P_G,c}$. For the general case of any $f$, let (abusing notation) $G = \text{GAN}_{\lambda f}(P_X, G; \mathcal{H}_c)$ and $\hat{G}$ denote the empirical counterpart with $n$ samples, and let $h^1, h^2 \in \mathcal{H}_c$ denote their witness functions. We then have

$G - \hat{G}$

$= \sup_{h \in \mathcal{H}_c} \left\{\mathbb{E}_{x \sim P_X}[h(x)] - \mathbb{E}_{x \sim P_G}[(\lambda f)^\star(h(x))]\right\} - \sup_{h \in \mathcal{H}_c} \left\{\mathbb{E}_{x \sim \hat{P}_X}[h(x)] - \mathbb{E}_{x \sim P_G}[(\lambda f)^\star(h(x))]\right\}$

$= \mathbb{E}_{x \sim P_X}[h^1(x)] - \mathbb{E}_{x \sim P_G}[(\lambda f)^\star(h^1(x))] - \mathbb{E}_{x \sim \hat{P}_X}[h^2(x)] + \mathbb{E}_{x \sim P_G}[(\lambda f)^\star(h^2(x))]$

$\leq \mathbb{E}_{x \sim P_X}[h^1(x)] - \mathbb{E}_{x \sim \hat{P}_X}[h^1(x)] + \mathbb{E}_{x \sim P_G}[(\lambda f)^\star(h^1(x))] - \mathbb{E}_{x \sim P_G}[(\lambda f)^\star(h^1(x))]$

$= \mathbb{E}_{x \sim P_X}[h^1(x)] - \mathbb{E}_{x \sim \hat{P}_X}[h^1(x)]$

$\leq \sup_{h \in \mathcal{H}_c} \left\{\mathbb{E}_{x \sim P_X}[h(x)] - \mathbb{E}_{x \sim \hat{P}_X}[h(x)]\right\}$

$= \text{IPM}_{\mathcal{H}_c}(P_X, \hat{P}_X)$

$\leq O(n^{-1/s_X}) + \frac{\Delta}{2}\sqrt{\frac{2}{n}\ln\left(\frac{1}{\delta}\right)},$

where the last step is an application of Lemma 21. Applying Theorem 8, we get $\hat{G} \leq \overline{W}_{c,\lambda \cdot f}$ and rearrangement of the above shows the first bound. For the case of $f(x) = |x - 1|$, note that if $\mathcal{F} \subseteq \mathscr{F}(\mathcal{X}, \mathbb{R})$ is such that $-\mathcal{F} = \mathcal{F}$, then $\text{IPM}_{\mathcal{F}}$ is a pseudo-metric and satisfies the triangle inequality, which allows us to have

$$\text{IPM}_{\mathcal{F}}(P_X, P_G) \leq \text{IPM}_{\mathcal{F}}(P_X, \hat{P}_X) + \text{IPM}_{\mathcal{F}}(\hat{P}_X, P_G)$$
$$\leq \text{IPM}_{\mathcal{F}}(P_X, \hat{P}_X) + \text{IPM}_{\mathcal{F}}(P_G, \hat{P}_G) + \text{IPM}_{\mathcal{F}}(\hat{P}_X, \hat{P}_G). \tag{17}$$

Next, we set $\mathcal{F} = \mathcal{F}_{c,\lambda}$, and noting that $\mathcal{F}_{c,\lambda} \subseteq \mathcal{H}_c$, we have

$$\text{IPM}_{\mathcal{F}_{c,\lambda}}(P_X, P_G) \leq \text{IPM}_{\mathcal{F}_{c,\lambda}}(P_X, \hat{P}_X) + \text{IPM}_{\mathcal{F}_{c,\lambda}}(P_G, \hat{P}_G) + \text{IPM}_{\mathcal{F}_{c,\lambda}}(\hat{P}_X, \hat{P}_G)$$
$$\leq \text{IPM}_{\mathcal{H}_c}(P_X, \hat{P}_X) + \text{IPM}_{\mathcal{H}_c}(P_G, \hat{P}_G) + \text{IPM}_{\mathcal{H}_c}(\hat{P}_X, \hat{P}_G)$$
$$\leq \text{IPM}_{\mathcal{H}_c}(\hat{P}_X, \hat{P}_G) + O(n^{-1/s_X} + n^{-1/s_G}) + \Delta\sqrt{\frac{2}{n}\ln\left(\frac{2}{\delta}\right)}, \tag{18}$$

where the final inequality is an application of Lemma 21 like before. However since we use McDiarmind's inequality twice, we set $\delta \leftarrow \delta/2$ and use union bound to have the above inequality with probability $1 - \delta$. The final step is to note that when $f(x) = |x - 1|$ then for any $\lambda > 0$,

$$(\lambda f)^\star(x) = \begin{cases} x & x \leq \lambda \\ \infty & x > \lambda \end{cases}$$

and so we have

$$\text{GAN}_{\lambda f}(P_X, G; \mathcal{H}_c) = \sup_{h \in \mathcal{H}_c} \{\mathbb{E}_{x \sim P_X}[h(x)] - \mathbb{E}_{x \sim P_G}[(\lambda f)^\star(h(x))]\}$$
$$= \sup_{h \in \mathcal{H}_c : |h| \leq \lambda} \{\mathbb{E}_{x \sim P_X}[h(x)] - \mathbb{E}_{x \sim P_G}[h(x)]\}$$
$$= \sup_{h \in \mathcal{F}_{c,\lambda}} \{\mathbb{E}_{x \sim P_X}[h(x)] - \mathbb{E}_{x \sim P_G}[h(x)]\}$$
$$= \text{IPM}_{\mathcal{F}_{c,\lambda}}(P_X, P_G).$$

By Theorem 8, we have $\text{IPM}_{\mathcal{F}_{c,\lambda}}(\hat{P}_X, \hat{P}_G) = \text{GAN}_{\lambda f}(\hat{P}_X, G; \mathcal{H}_c) \leq \overline{W}_{c,\lambda \cdot f}(\hat{P}_X, G)$ where $\text{GAN}_{\lambda f}(\hat{P}_X, G; \mathcal{H}_c)$ is the objective with $\hat{P}_X$ and $\hat{P}_G$. Putting this together with (18), we get

$$\text{GAN}_{\lambda f}(P_X, G; \mathcal{H}_c) = \text{IPM}_{\mathcal{F}_{c,\lambda}}(P_X, P_G)$$

$$\leq \text{IPM}_{\mathcal{H}_c}(\hat{P}_X, \hat{P}_G) + O(n^{-1/s}) + \Delta\sqrt{\frac{2}{n}\ln\left(\frac{1}{\delta}\right)}$$

$$= \text{GAN}_{\lambda f}(\hat{P}_X, G; \mathcal{H}_c) + O(n^{-1/s}) + \Delta\sqrt{\frac{2}{n}\ln\left(\frac{1}{\delta}\right)}$$

$$\leq \overline{W}_{c,\lambda \cdot f}(\hat{P}_X, G) + O(n^{-1/s_X} + n^{-1/s_G}) + \Delta\sqrt{\frac{2}{n}\ln\left(\frac{2}{\delta}\right)}.$$

### A.3 Proof of Theorem 9

First, using Theorem 8 and the fact that the $f$-GAN objective is a lower bound to $D_f$, we have that

$$\overline{W}_{\gamma \cdot c, f}(P_X, G) = \text{GAN}_f(P_X, G, \mathcal{H}_{\gamma c})$$
$$\leq D_f.$$

It is known that $f'(dP_X/dP_G)$ is the maximizer of $L(h) = \mathbb{E}_{x \sim P_X}[h(x)] - \mathbb{E}_{x \sim P_G}[f^\star(h(x))]$ [15], and so the proof concludes by showing that $f'(dP_X/dP_G) \in \mathcal{H}_{\gamma^\star \cdot c}$. Note that $h \in \mathcal{H}_{\gamma \cdot c}$ if and only if for all $x, x' \in \mathcal{X}, x \neq x'$

$$|h(x) - h(x')| \leq \gamma \cdot \delta_{x-x'}(0)$$
$$= \gamma$$

and so the 1-Lipschitz functions are those that are bounded by their maximum and minimum value by $\gamma$. For any $x, x' \in \mathcal{X}, x \neq x'$ we have

$$\left| f'\left(\frac{dP_X}{dP_G}\right)(x) - f'\left(\frac{dP_X}{dP_G}\right)(x') \right| = \gamma^* \left| f'\left(\frac{dP_X}{dP_G}\right)(x) - f'(0) \right|$$
$$\leq \gamma,$$

and thus $f'(dP_X/dP_G) \in \mathcal{H}_{\gamma \cdot c}$.

## A.4 Proof of Theorem 10

First note that

$$\text{WAE}_{c, \lambda \cdot f}(P_X, P_G) = \inf_{E \in \mathcal{F}(\mathcal{X}, \mathscr{P}(\mathcal{Z}))} \left\{ \int_{\mathcal{X}} \mathbb{E}_{z \sim E(x)}[c(x, G(z))]dP_X(x) + \lambda \cdot D_f(E \# P_X, P_Z) \right\}$$

$$\leq \inf_{E \in \mathcal{F}(\mathcal{X}, \mathscr{P}(\mathcal{Z})): E \# P_X = P_Z} \left\{ \int_{\mathcal{X}} \mathbb{E}_{z \sim E(x)}[c(x, G(z))]dP_X(x) \right\}$$

$$= W_c(P_X, P_G),$$

where the last equality holds from [6] Theorem 1. Thus we have the chain of inequalities for all $\lambda$ and $f : \mathbb{R} \to (-\infty, \infty]$ (convex with $f(1) = 0$)

$$\text{GAN}_{\lambda f}(P_X, G; \mathcal{H}_c) \leq \overline{W}_{c, \lambda \cdot}(P_X, P_G) \leq \text{WAE}_{c, \lambda \cdot f}(P_X, P_G) \leq W_c(P_X, P_G).$$

We now show the opposite direction, which will conclude the proof.

**Lemma 23** *For any metric $c$ and $f : \mathbb{R} \to (-\infty, \infty]$ convex function with $f(1) = 0$, if*
$$\lambda \geq \lambda^* = \sup_{P' \in \mathscr{P}(\mathcal{X})} \left( W_c(P', P_G)/D_f(P', P_G) \right),$$

*then we have*

$$\text{GAN}_{\lambda f}(P_X, G; \mathcal{H}_c) \geq W_c(P_X, P_G)$$

**Proof** First noting that $\lambda \geq \sup_{P' \in \mathscr{P}(\mathcal{X})} \left( W_c(P', P_G)/D_f(P', P_G) \right)$, for all $P' \in \mathscr{P}(\mathcal{X})$, we have
$$\lambda D_f(P', P_G) - W_c(P', P_G) \geq 0.$$

Let $\tilde{\mathcal{Z}} = \mathcal{X}, \tilde{G} = \text{Id}, P_{\tilde{\mathcal{Z}}} = P_G$ and noting that $\tilde{G}$ is invertible, we can apply Theorem 8 to get

$$\text{GAN}_{\lambda f}(P_X, G; \mathcal{H}_c) = \overline{W}_{c, \lambda \cdot f}(P_X, \tilde{G} \# P_{\tilde{\mathcal{Z}}})$$
$$= \inf_{E \in \mathcal{F}(\mathcal{X}, \mathscr{P}(\mathcal{X}))} \{ W_c(E \# P_X, P_X) + \lambda D_f(E \# P_X, P_G) \}$$
$$\geq \inf_{E \in \mathcal{F}(\mathcal{X}, \mathscr{P}(\mathcal{X}))} \{ W_c(P_X, P_G) - W_c(E \# P_X, P_G) + \lambda D_f(E \# P_X, P_G) \}$$
$$\geq \inf_{E \in \mathcal{F}(\mathcal{X}, \mathscr{P}(\mathcal{X}))} \{ W_c(P_X, P_G) \}$$
$$= W_c(P_X, P_G). \qquad \blacksquare$$

## A.5 Proof of Theorem 14

We have

$$\overline{W}_{c, \lambda \cdot f}(P_X, G) = \inf_{E \in \mathcal{F}(\mathcal{X}, \mathscr{P}(\mathcal{Z}))} \{ W_c(P_X, (G \circ E) \# P_X) + \lambda D_f(E \# P_X, P_Z) \}$$

$$\leq \inf_{E \in \mathcal{F}(\mathcal{X}, \mathscr{P}(\mathcal{Z})): E \# P_X = P_Z} \{ W_c(P_X, (G \circ E) \# P_X) + \lambda D_f(E \# P_X, P_Z) \}$$

$$= \inf_{E \in \mathcal{F}(\mathcal{X}, \mathscr{P}(\mathcal{Z})): E \# P_X = P_Z} \{ W_c(P_X, (G \circ E) \# P_X) \}$$

$$= \inf_{E \in \mathcal{F}(\mathcal{X}, \mathscr{P}(\mathcal{Z})): E \# P_X = P_Z} \{ W_c(P_X, P_G) \}$$

$$= W_c(P_X, P_G).$$