[Reviews · NeurIPS 2019]

Reviewer 1



Summary: this paper investigates the primal-dual link between f-GANs and Wasserstein autoencoders (WAEs). Originality: this paper provides many theoretical results concerning generative methods. Even though I am not very familiar with the theory of optimal transport, the proposed content seems novel, in particular in the field of GANs and autoencoders. A generalization bound for these methods is also provided, as a complete novelty. Quality: I was not able to check all the proofs but the paper seems technically sound. I wonder in which sens Theorem 13 is a generalization bound. Clarity: Overall, the paper is not very easy to read. The theoretical results are claimed without clear explanations. Why are the results important and useful? As minor remark, it is unclear how the rhs of (2) depends on X and Z. Significance: this analysis that is conducted by this paper is useful to understand the good and similar performances of f-GANs and WAEs. Rebuttal: I have read the authors' response and I thank them for clarifying their work.

Reviewer 2



Summary: It is the purpose of this article to establish links between the two most popular generative models nowadays, namely GANs and (W)AEs. In Theorem 8, an (in)equality linking both criteria is stated, tending to explain the performance similarities between the models. An introduced f-WAE model is also thoroughly analyzed, linked to an f-divergence or a Wasserstein distance depending on the weighting operated. Finally, Authors use findings of Theorem 8 to derive generalization bounds on WAEs. Major Comments: - The paper is well written, related works are correctly discussed and notions well introduced, making the submission self-contained although quite dense. - The established results are strong and profound, making them of particular interest on such a hot topic as generative models. The fact that Theorem 8 can be viewed as a generalization of Lemma 4 is particularly striking, and attest of the soundness of the approach. - Generalization bounds derivations are an attractive "application" of the previously derived equivalence, endorsing the utility of a unifying framework. - Authors mentioned the "contrasting abilities" of GANs and WAEs, can the derived equivalence be used to design a "best of the two world" training strategy, attaining high performances while avoiding instability? Minor Comments: - l. 116: a precision on the reference (Lemma/Theorem number in [14]) may be interesting - l. 215: the same objective*s* - l. 242: it was show *that* - l. 245: a prac*ti*tioner Overall Evaluation: The theoretical results seem profound and their derivations are well conducted, although I may lack knowledge on the related works to fully asses it. It is a good submission.

Reviewer 3



The paper is sufficiently original for presentation at NeurIPS. It draws very interesting conclusions from a very relevant duality based approach to the analysis of GAN's. The analysis is technically sound and reading through the proofs gave me the impression that they are correct. The clarity of the exposition is of a sufficiently high level. The results are very significant and fill a gap in the current knowledge about GANs and similar generative approaches. It will help the community reaching a better understanding of this type of techniques and their relationships.

[Author Response · NeurIPS 2019]

R1 - Thank you for your review and your comments regarding the clarity of our results. Theorem 13 is a generalization bound in the sense that we define a notion of generalization: quantifying the difference between the true distribution $P_X$ and $P_G$ and present a bound on this quantity (hence "generalization bound"). We will clarify these theoretical results, the dependence of X and Z of eqn (2) and check the references you mentioned – thank you for pointing this out.

R2 - Thank you for your review and comments regarding the profoundness of our results. We will be addressing the minor remarks mentioned. Though our result mainly discovers the relationship between these methods, one remark towards suggesting a direction is the use of lipschitz discriminators for GANs which lead to stable performance closer to WAEs – thank you for pointing this out for our discussion.

R3 - Thank you for your review and pointing out the significance of our results and especially noting that it will help the community generally understand these methods better. Indeed, cases under which the equivalence does not hold are interesting, and so we will endeavor to add appropriate illustrations for such cases (probably in an appendix given the space constraint in the main body).

[Meta-Review · NeurIPS 2019]

The contribution of the paper has been judged significant, theoretically sound and rather well-written. For this reason, acceptance is recommended.